# Out-of-Distribution Generalized Graph Anomaly Detection with Homophily-aware Environment Mixup

**Sibo Tian[1], Xin Wang[1]**\*, **Zeyang Zhang[1], Haibo Chen[1], Wenwu Zhu[1]**\*
[1]Tsinghua University

## Abstract

Graph anomaly detection (GAD) is widely prevalent in scenarios such as financial fraud detection, anti-money laundering and social bot detecion. However, structural distribution shifts are commonly observed in real-world GAD data due to selection bias, resulting in reduced homophily. Existing GAD methods tend to rely on homophilic shortcuts when trained on high-homophily structures, limiting their ability to generalize well to data with low homophily under structural distribution shifts. In this study, we propose to handle structural distribution shifts by generating novel environments characterized by diverse homophilic structures and utilizing invariant patterns, *i.e.*, features and structures with the capability of stable prediction across structural distribution shifts, which face two challenges: (1) How to discover invariant patterns from entangled features and structures, as structures are sensitive to varying homophilic distributions. (2) How to systematically construct new environments with diverse homophilic structures. To address these challenges, we propose Ego-Neighborhood Disentangled Encoder with **H**omophily-aware **E**nvironment **M**ixup (**HEM**), which effectively handles structural distribution shifts in GAD by discovering invariant patterns. Specifically, we first propose an ego-neighborhood disentangled encoder to decouple the learning of feature and structural embeddings, which facilitates subsequent improvements in the invariance of structural embeddings for prediction. Next, we introduce a homophily-aware environment mixup that dynamically adjusts edge weights through adversarial learning, effectively generating environments with diverse structural distributions. Finally, we iteratively train the classifier and environment mixup via adversarial training, simultaneously improving the diversity of constructed environments and discovering invariant patterns under structural distribution shifts. Extensive experiments on real-world datasets demonstrate that our method outperforms existing baselines and achieves state-of-the-art performance under structural distribution shift.

## 1 Introduction

Graph anomaly detection (GAD) [1] represents a classical task in graph machine learning, with extensive applications in financial fraud detection[2], anti-money laundering[3], and social bot detection[4, 5]. Typically, GAD can be formulated as a semi-supervised node classification problem[6], where the objective is to identify anomalous nodes in the input graph. Graph neural networks (GNNs) have demonstrated superior predictive capabilities in GAD tasks due to their ability to handle complex topological relationships and model sophisticated node representations[7].

However, structural distribution shifts[8–11] are prevalent in real-world graph anomaly detection scenarios due to factors such as selection bias[12]. For example, in anti-money laundering contexts[13], coordinated money laundering accounts are more likely to be identified and labeled compared to

---

\*Corresponding authors. {xin_wang,wwzhu}@tsinghua.edu.cn

isolated ones, while large-scale coordinated bot comments on social platforms are more easily detected by automated systems. This results in a structural distribution shift between training and test data, as nodes with stronger homophily and more connections to similar nodes are more likely to be labeled. Existing GNNs for GAD methods trained on high-homophily structures tend to rely on the homophilic shortcut, limiting their ability to generalize effectively to test data with low homophily under structural distribution shifts.

In this paper, we study the problem of handling structural distribution shifts in graph anomaly tasks by generating novel environments characterized by diverse homophilic structures and utilizing invariant patterns, *i.e.*, features and structures with the capability of stable prediction across structural distribution shifts, which remains largely unexplored in the literature. However, this problem is highly nontrivial with the following challenges:

- How to discover invariant patterns from entangled features and structures, as structures are sensitive to varying homophilic distributions.
- How to systematically construct new environments with diverse homophilic structures.

To address these challenges, we propose a novel framework named Ego-Neighborhood Disentangled Encoder with **H**omophily-aware **E**nvironment **M**ixup (**HEM**) to discover invariant patterns with stable predictive abilities under structural distribution shifts. Specifically, we first propose an ego-neighborhood disentangled encoder to separately model the feature and structure representations. By this design, we can (1) disentangle feature information and structure information, enabling the following modules to strengthen the invariance of the prediction patterns. (2) model the interaction patterns between ego-node and neighborhood to better predict the anomaly probability. Then, we propose a homophily-aware environment mixup, which dynamically adjusts edge weights within the graph to implicitly modify the structure of ego graphs. This mechanism generates diverse training environments and facilitates the learning of invariant patterns in a memory-efficient manner. Finally, the two modules are trained iteratively in an adversarial manner, in order to improve the diversity of the constructed environment and discover invariant patterns under structural distribution shifts. Extensive experiments on real-world datasets demonstrate that our proposed method achieves state-of-the-art performance in GAD under structural distribution shift scenarios.

The contributions of our work can be summarized as follows:

- We propose Ego-Neighborhood Disentangled Encoder with **H**omophily-aware **E**nvironment **M**ixup (**HEM**), which can handle structural distribution shift in graph anomaly detection, which is largely unexplored in literature.
- We propose an ego-neighborhood disentangled encoder to disentangle feature and structure representations for learning invariant patterns, which is a general framework that can be utilized to improve model performance for many GNNs. Besides, we propose a homophily-aware environment mixup to efficiently generate training environments with diverse local structures to improve the model's generalization capability.
- Experiments on real-world datasets demonstrate that our method achieves state-of-the-art performance compared to existing baselines.

## 2   Problem Formulation

In this section, we formulate the problem of graph anomaly detection under distribution shift.

**Node-Attribute Graph**    A node-attribute graph G consists of a node set V, an edge set E, and node attributes $X \in R^{n \times d}$, where each row of X represents the feature vector of the corresponding node. The edges between nodes are represented by the adjacency matrix $A \in R^{n \times n}$, where n is the number of nodes and d is the dimension of the input features. The node labels are defined as $Y \in R^{n \times 1}$. Thus, a node-attribute graph can be summarized as $G = \{A, X, Y\}$. In node-level graph anomaly detection, the labels are binary variables indicating whether a node is normal or anomalous.

**Node-Level Graph Anomaly Detection (GAD)**    Node-level graph anomaly detection is an imbalanced binary classification task. The goal of the proposed model is to predict whether a node is normal or anomalous. In this paper, we focus on the transductive setting, while our proposed

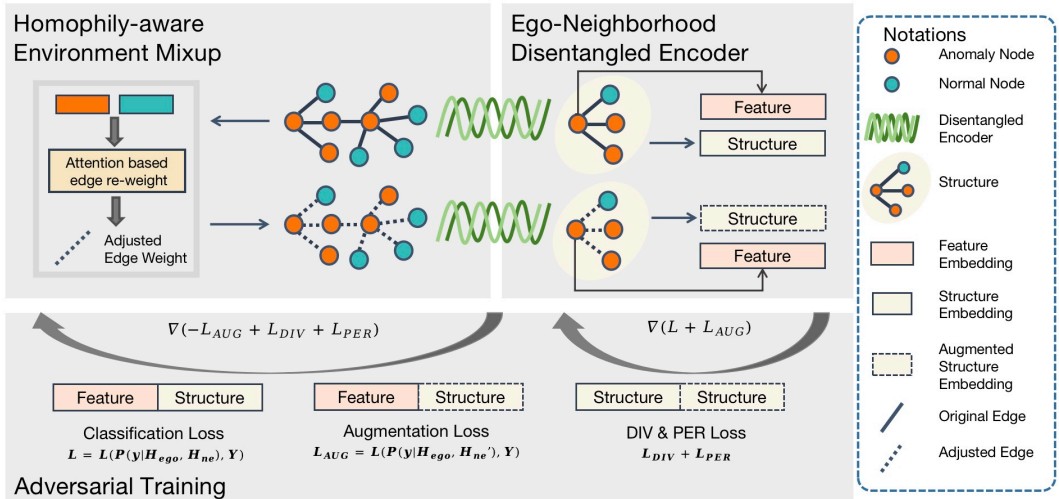

Figure 1: The overall framework of our proposed **HEM**. The framework transforms an original graph into an augmented graph by dynamically adjusting edge weights with homophily-aware environment mixup. Both graphs are processed by the ego-neighborhood disentangled encoder to extract feature and structural representations. The original loss $L$ and augmented loss $L_{AUG}$ are computed using cross-entropy with the disentangled representations, while the diversity loss $L_{DIV}$ measures the difference between original and augmented structural embeddings, and the PER loss $L_{PER}$ constrains the proportion of perturbed edges to avoid trivial solutions. These losses jointly optimize the encoder and environment mixup, enhancing generalization across varying structural distributions.

method can be simply applied in inductive scenarios. Following [14] [15] [16] [17], we formulate this problem as predicting the probability to be an anomaly of a node given its induced ego-graph, i.e., $p(Y_v|G_v)$. A k-order ego-graph $G_v$ induced from v is a subgraph containing node v with all of its k-hop neighbors, k is an arbitrary integer. The optimization objective is to learn an optimal predictor to model the node's anomaly probability

$$\min_{\theta} \mathbb{E}_{(y_v, G_v) \sim p(\mathbf{y_v}, \mathbf{G_v})} L(f_\theta(G_v), y_v) \tag{1}$$

where $f_\theta$ is a graph neural network parameterized by $\theta$. Denote random variable of ego-graph and node label as $G_v, y_v$, and respective instances as $\mathbf{G_v}, \mathbf{y_v}$.

**Structural Distribution Shift**    Structural distribution shift in node-level tasks can be treated as a covariate shift, which means the ego-graph distribution is different across training and test data, i.e. $p_{train}(\mathbf{G_v}) \neq p_{test}(\mathbf{G_v})$, leading to different local structures. Therefore, we formulate this problem as an out-of-distribution (OOD) problem,

$$\min_{\theta} \max_{e \in \mathcal{E}} \mathbb{E}_{(y_v, G_v) \sim p(\mathbf{y_v}, \mathbf{G_v}|\mathbf{e}=e)} [L(f_\theta(G_v), y_v)|e]. \tag{2}$$

In GAD tasks, structural distribution shift is a common problem[12], due to factors like sampling bias. For example, in e-commerce networks, accounts that transact with anomalous accounts are more likely to be anomalous themselves, potentially indicating participation in money laundering transactions. This will cause a structural distribution shift that ego-graphs in the training set show a higher homophily ratio, i.e. the proportion of neighbors sharing the same label as the central node is much higher than that in the test set. The homophily ratio is defined as $\text{Homo}(v) = \frac{1}{|N(v)|}|u \in N(v)|Y_u = Y_v|$. The ego-graphs with a high homophily ratio in the training stage may lead GNNs to rely on the homophilic shortcut, a spurious pattern that predicts nodes' labels by the majority label in their neighborhood, and finally to fail in the test stage ego-graphs with low homophily ratio.

# 3 Method

In this section, we propose the **HEM**, a novel framework for graph anomaly detection under structural distribution shift by introducing two key modules, ego-neighborhood disentangled encoder and homophily-aware environment mixup.

## 3.1 Ego-Neighborhood Disentangled Encoder

It's critical to utilize both knowledge from ego-node and neighborhood to extract invariant patterns in graph anomaly detection for two reasons. (1) The interaction between the ego node and neighborhood indicates the discrepancy, which is important for predicting anomalies. (2) Excessive reliance on structural information results in unstable predictions under structural distribution shifts. Encoding both feature and structure information with one encoder makes the variant and invariant patterns entangled, leading to a performance drop under distribution shift. Most existing works simply employ a single GNN as an encoder, leading to fail in test data with distribution shift. While CoLA[15] models anomaly probability via local discrepancy, its method faces a trade-off between sampling size and speed, resulting in suboptimal performance.

To overcome these limits, we propose an ego-neighborhood disentangled encoder to efficiently and effectively extract feature and structural patterns from ego graphs in a decoupled manner.

**Neighborhood Encoder**  We define the $k$-hop neighborhood as the ego-graph for a given node. The neighborhood encoder can be implemented using any commonly used message-passing-based GNN architecture. In our framework, we adopt the BWGNN[18], a strong backbone in GAD tasks, as the neighborhood encoder.

Given the initial node features X and the adjacency matrix $A$, we first employ an Multi-Layer Perceptron (MLP) projector to project it into the embedding space.

$$H_0 = \text{Proj}(X) \tag{3}$$

we can generally compute the neighborhood embeddings as follows

$$H_{ne} = GNN(H_0, A) \tag{4}$$

Specifically, with BWGNN applied, the neighborhood embeddings can be modeled as

$$H_{ne} = \sum_{k=0}^{K} w_k \hat{A}^k H_0 \tag{5}$$

where $\hat{A}$ is the normalized adjacency matrix, i.e. $\hat{A} = D^{-\frac{1}{2}} A D^{-\frac{1}{2}}$, D is the diagonal matrix. $w_k$ is the k th coefficient from a Beta polynomial.

**Ego-Node Encoder**  The relationship between ego-node's feature and label is stable under structural distribution shift since ego-nodes only contain information about a single instance. We disentangle them from structural patterns so that we can model the anomaly probability by the interaction patterns. Specifically, we employ an MLP to encode the ego-node's representation. To align the neighborhood embedding space and ego-node embedding space, we utilize the same projector as the neighborhood embedding projector

$$H_{ego} = H_0 = \text{Proj}(X). \tag{6}$$

The sharing projector ensures that the embeddings generated by the two encoders are aligned within the same embedding space, facilitating effective interaction between the ego node and its neighborhood embeddings.

**Contrastive Prediction**  To capture the discrepancy between the ego-node embedding and structural embedding, we employ a bilinear function to predict anomaly probability. This design is commonly employed by contrastive learning methods [15, 19].

$$P(y \mid h_{\text{ego}}, h_{\text{ne}}) = \sigma(h_{\text{ego}}^T W h_{\text{ne}}), \tag{7}$$

where $\sigma$ denotes the sigmoid function, and $W$ is a learnable weight matrix that models the interaction between the ego node and structural embeddings.

The classification loss can be computed as

$$L = \mathcal{L}(P(y \mid H_{\text{ego}}, H_{\text{ne}}), Y) \tag{8}$$

where $\mathcal{L}$ is cross-entropy loss.

## 3.2 Homophily-aware Environment Mixup

Existing out-of-distribution graph anomaly detection methods suffer from issues like reliance on labeled and diverse environments[20, 21], high memory cost[14], or limited performance due to lack of supervision[22].

To overcome these limitations, we propose a homophily-aware environment mixup, which generates diverse environments exhibiting effective structural distribution shifts by dynamically re-weighting edges based on the homophily observed among neighboring nodes. This process is both computationally and memory efficient.

**Homophily-aware environment mixup**  We propose a homophily-aware environment mixup, to generate environments with structural distribution shift and improve the generalization capability of the classifier across environments with different local structures. Our objective for this module is to generate new environments with diverse local structures which is challenging enough to the classifier. This can be formulated as

$$\max_{e \in \mathcal{E}} \mathbb{E}_{(y_v, G_v) \sim p(\mathbf{y_v}, \mathbf{G_v} \mid \mathbf{e}=e)}[L(f_\theta(G_v), y_v)|e]. \tag{9}$$

As mentioned above, homophily is an important metric to describe local structure. An obvious shift between training and test environments is the significant drop in homophily ratio. Motivated by this, we propose an attention mechanism that dynamically adjusts edge weights according to the features of the nodes connected by the edge. Specifically, following [23], we employ a node-wise attention mechanism to model the connection strength between the nodes connected by the same edge as follows

$$w_{(u,v)} = \text{sigmoid}(\text{LeakyReLU}(\text{concat}(h_{ego|u}, h_{ego|v})W)), \tag{10}$$

where $W$ is a learnable attention weight matrix. $w_{(u,v)}$ is the adjusted weight, normalized to [0, 1]. Then we apply the disentangled encoder on the generated graph with the adjusted edge weights in the message passing process, resulting in ego-node and neighborhood embeddings $H'_{\text{ego}}$ and $H'_{\text{ne}}$. The augmented loss is defined as:

$$L_{AUG} = \mathcal{L}(P(y \mid H'_{\text{ego}}, H'_{\text{ne}}), Y) \tag{11}$$

Our design offers three key advantages. First, in contrast to methods that directly add or drop edges, our approach enables continuous modulation of edge weights, allowing for a more fine-grained perturbation of local structures. This continuous adjustment effectively mixes the discrete states of connected and disconnected edges, providing a richer spectrum of graph augmentations. Second, by optimizing the environment generator rather than directly manipulating several fully connected adjacency matrices, our method achieves significant memory efficiency. This approach circumvents the substantial memory overhead associated with storing and processing dense adjacency representations in EERM[14]. Third, our method leverages the features of both endpoints of an edge to perform homophily-aware edge weight adjustments, as opposed to random or uninformed modifications, leading to generating environments with diverse local structures with respect to homophily.

**Diversity Loss**  To ensure the module generates diverse local structures, we introduce a diversity loss that maximizes the dissimilarity between the original neighborhood embedding and the generated neighborhood embedding:

$$L_{DIV} = \text{Cosine Similarity}(H_{ne}, H'_{ne}), \tag{12}$$

where $H_{subgraph}$ and $H'_{subgraph}$ represent the embeddings of the original and augmented subgraphs, respectively.

**Preserved Edge Ratio(PER) Loss**   To prevent trivial solutions, such as adjusting all edge weights to zero, we regularize the environment generator to perturb only a small subset of edges. This is achieved by minimizing the following PER loss:

$$L_{PER} = \left\| \frac{\sum_{e \in E} w(e)}{|E|} - \rho \right\|_2,$$  (13)

where $\rho$ is the target proportion of edges to preserve from perturbation.

## 3.3   Overall Adversarial Training

To jointly optimize the disentangled encoder and the environment mixup, we employ an iterative adversarial training framework. This framework consists of an outer loop for training the encoder and an inner loop for training the environment mixup, ensuring that both components are optimized in a coordinated manner.

**Outer Loop: Training the Disentangled Encoder**   In the outer loop, we train the encoder using a composite loss function that mixes the standard classification loss $L$ and the augmented loss $L_{AUG}$. The augmented loss $L_{AUG}$ ensures that the model generalizes well to environments with structural distribution shifts. The overall loss function for the outer loop is defined as:

$$L_{\text{outer}} = L + L_{AUG}.$$  (14)

**Inner Loop: Training the Environment Mixup**   In the inner loop, we optimize the environment mixup to generate diverse and challenging augmented environments. The corresponding loss function combines the augmented loss $-L_{AUG}$, the diversity loss $L_{DIV}$, and the PER loss $L_{PER}$. Specifically:

$$L_{\text{inner}} = -L_{AUG} + L_{DIV} + L_{PER}.$$  (15)

Here, $-L_{AUG}$ encourages to generate more challenging environments, $L_{DIV}$ maximizes the dissimilarity between original and generated environments to enhance diversity, and $L_{PER}$ ensures that only a small subset of edges is perturbed, avoiding trivial solutions.

**Training Dynamics**   The training process alternates between the outer and inner loops, as shown in the Algorithm. 1. In each iteration of the outer loop, the encoder is updated to minimize $L_{\text{outer}}$, while in the inner loop, the environment mixup is updated to minimize $L_{\text{inner}}$. This adversarial training strategy encourages to generation of new distributions with diverse local structures and to improve the generalization capability across different distributions.

---

**Algorithm 1** Training pipeline for **HEM**

---
**Require:** Training epochs $L$, edge preservation ratio $\rho$
1: **for** $l = 1, \ldots, L$ **do**
2:     Obtain $H_{ego}$, $H_{ne}$ for each node as described in Section 3.1
3:     Calculate classification loss $L$ as Eq. 8
4:     Generate new environment and calculate classification loss in generated environment $L_{AUG}$
        as Eq. 11
5:     Calculate diversity loss $L_{DIV}$ and PER loss $L_{PER}$ as Eq. 12 and Eq. 13
6:     Calculate inner loss $L_{inner}$ and outer loss $L_{outer}$ as Eq. 15 and Eq. 14
7:     Update the homophily-aware environment mixup by minimizing inner loss
8:     Update the disentagled encoder by minimizing outer loss
9: **end for**

---

## 3.4   Theoretical Explanation

To theoretically justify the proposed **HEM** framework, we analyze how it promotes prediction stability and invariant learning across varying homophily ratios through a simplified example.

**Variable Definitions.** Let $h$ denote the homophily ratio and $Y \in \{0, 1\}$ denote the node label, where $Y = 1$ indicates an anomalous node and $Y = 0$ a normal node. Let $\pi = P(Y = 1)$ be the anomaly rate, and denote by $r_v$ the proportion of anomalous neighbors for node $v$:

$$r_v = \frac{1}{d_v} \sum_{u \in \mathcal{N}(v)} Y_u, \tag{16}$$

where $d_v = |\mathcal{N}(v)|$ is the node degree. We assume a constant node degree d. Let $\mu_1 = \mathbb{E}[X \mid Y = 1]$ and $\mu_0 = \mathbb{E}[X \mid Y = 0]$, and denote $\Delta\mu = \mu_1 - \mu_0 \neq 0$. For a one-layer message-passing encoder followed by a linear classifier, the node embedding can be expressed as

$$Z_v = r_v\mu_1 + (1 - r_v)\mu_0 = \mu_0 + r_v\Delta\mu. \tag{17}$$

**Lemma 1 (Predictor Dependence on Homophily).** The optimal coefficient that minimizes the Mean Squared Error (MSE) between predictions and labels is

$$\alpha^*(h) = \frac{\mathrm{Cov}(Y, r)}{\mathrm{Var}(r)} = \frac{\pi(1 - \pi)(2h - 1)}{\frac{h(1-h)}{d} + \pi(1 - \pi)(2h - 1)^2}. \tag{18}$$

Hence, the learned predictor $\alpha^*$ depends explicitly on the homophily ratio $h$; a model trained under homophily $h_{\mathrm{tr}}$ will be suboptimal for a test distribution with a different $h_{\mathrm{te}}$.

**Lemma 2 (Bounded Deviation on the Test Distribution).** If there exists an $\epsilon$ such that $|h_{\mathrm{te}} - h_{\mathrm{tr}}| \leq \epsilon$, with the proposed environment mixup and adversarial training, the gap between the expected test loss of the training-optimal parameters and the optimal test loss is bounded. Specifically,

$$L(h_{\mathrm{te}}, \alpha^*_{\mathrm{tr}}) - L(h_{\mathrm{te}}, \alpha^*_{\mathrm{te}}) \leq V_{\max}L_f\epsilon^2 \tag{19}$$

, where $L_f$ denotes the Lipschitz constant of $f(h) = \alpha^*_h$, and $V_{\max}$ represents the maximum of $Var(r)$ for $h$ satisfying $|h - h_{\mathrm{tr}}| \leq \epsilon$.

# 4 Experiments

In this section, we conduct various experiments to verify that our proposed method can handle structural distribution shifts in graph anomaly detection tasks by utilizing the invariant patterns in the disentangled subgraph representations.

## 4.1 Datasets

We choose 3 commonly used GAD datasets, including Amazon, Yelp, and T-finance. We provide the details of the datasets in Appendix A.

## 4.2 Baselines

To demonstrate the effectiveness of our proposed method in addressing structural distribution shifts in graph anomaly detection tasks through a complete comparison, we have selected four types of baselines for evaluation, including (1) General GNNs: GCN[24], GAT[23], GraphSAGE[25], (2)GNNs Specialized for Graph Anomaly Detection: BernNet[26], BWGNN[18], GHRN[27], PCGNN[28], BAT[29](3)General Out-of-Distribution Methods: GroupDRO[21], V-REx[20], (4)Graph Out-of-Distribution Methods: SRGNN[22] , EERM[14], GDN[12].Details of these baseline models are provided in Appendix B.

## 4.3 Settings

We adopt the above three real-world datasets as node-level graph anomaly detection tasks. We transform Amazon and Yelp into homogeneous graphs by simply merging all types of edges into one type to make a comparison between homogeneous GNNs and heterogeneous GNNs. We divide each dataset into 2 domains, with/without structural distribution shift. For brevity, we denote domains with/without distribution shift as 'w/ DS' and 'w/o DS' separately. Specifically, we sample nodes by their homophily ratio, which means nodes with low homophily are more likely to be divided

Table 1: Results of different methods on real-world graph anomaly detection datasets. The best results are in bold and the second-best results are underlined. The results are reported in AUPRC.

| Dataset | Amazon | | Yelp | | T-Finance | |
|---|---|---|---|---|---|---|
| Model | w/o DS | w/ DS | w/o DS | w/ DS | w/o DS | w/ DS |
| GCN | **94.45**$_{\pm0.97}$ | 69.90$_{\pm2.13}$ | 53.63$_{\pm0.94}$ | 40.79$_{\pm1.73}$ | 93.66$_{\pm0.10}$ | 60.33$_{\pm0.41}$ |
| GAT | 92.84$_{\pm0.00}$ | 72.44$_{\pm0.00}$ | 54.74$_{\pm0.00}$ | 42.83$_{\pm0.00}$ | 87.38$_{\pm0.00}$ | 53.93$_{\pm0.00}$ |
| GraphSAGE | 90.57$_{\pm1.03}$ | 69.71$_{\pm1.00}$ | **63.05**$_{\pm0.24}$ | 53.25$_{\pm0.76}$ | 73.61$_{\pm0.47}$ | 44.21$_{\pm0.98}$ |
| BernNet | 92.83$_{\pm0.02}$ | 71.34$_{\pm0.21}$ | 52.99$_{\pm0.20}$ | 48.53$_{\pm0.32}$ | 89.25$_{\pm0.71}$ | 65.21$_{\pm0.58}$ |
| PCGNN | 90.95$_{\pm0.01}$ | 72.02$_{\pm0.03}$ | 54.05$_{\pm0.78}$ | 44.52$_{\pm0.32}$ | 69.37$_{\pm1.54}$ | 36.84$_{\pm2.07}$ |
| GHRN | 92.01$_{\pm0.25}$ | 69.39$_{\pm0.14}$ | 59.27$_{\pm0.45}$ | 54.68$_{\pm0.61}$ | 93.62$_{\pm0.15}$ | 62.43$_{\pm0.36}$ |
| BWGNN | 91.59$_{\pm0.20}$ | 69.26$_{\pm0.38}$ | 58.21$_{\pm0.32}$ | 52.97$_{\pm0.51}$ | 93.44$_{\pm0.10}$ | 62.34$_{\pm0.72}$ |
| BAT | 94.03$_{\pm2.04}$ | 72.26$_{\pm1.16}$ | 62.19$_{\pm2.61}$ | 54.93$_{\pm0.49}$ | 92.77$_{\pm0.67}$ | 56.35$_{\pm1.75}$ |
| V-REx | 92.54$_{\pm0.14}$ | 68.84$_{\pm0.23}$ | 59.16$_{\pm0.70}$ | 53.58$_{\pm0.65}$ | 93.04$_{\pm0.12}$ | 60.66$_{\pm0.78}$ |
| GroupDRO | 92.58$_{\pm0.04}$ | 68.87$_{\pm0.29}$ | 58.73$_{\pm0.66}$ | 52.13$_{\pm0.31}$ | 93.86$_{\pm0.13}$ | 60.81$_{\pm0.28}$ |
| SRGNN | 90.48$_{\pm0.36}$ | 70.36$_{\pm0.56}$ | 58.73$_{\pm0.21}$ | 52.84$_{\pm0.40}$ | 91.11$_{\pm0.24}$ | 60.36$_{\pm0.14}$ |
| EERM | 92.57$_{\pm0.01}$ | 70.94$_{\pm0.51}$ | OOM | OOM | OOM | OOM |
| GDN | 91.97$_{\pm0.50}$ | 68.93$_{\pm0.43}$ | 48.20$_{\pm1.13}$ | 48.43$_{\pm0.92}$ | 92.51$_{\pm0.07}$ | 59.84$_{\pm0.05}$ |
| **HEM(+GCN)** | 93.70$_{\pm0.11}$ | 70.59$_{\pm0.09}$ | 55.09$_{\pm0.47}$ | 52.05$_{\pm1.00}$ | **95.69**$_{\pm0.08}$ | 64.97$_{\pm0.11}$ |
| **HEM(+BWGNN)** | 91.97$_{\pm0.74}$ | **74.91**$_{\pm0.49}$ | 61.42$_{\pm0.56}$ | **58.45**$_{\pm1.01}$ | 94.76$_{\pm0.06}$ | **66.04**$_{\pm0.29}$ |

Table 2: Results of different methods on real-world graph anomaly detection datasets. The best results are in bold and the second-best results are underlined. The results are reported in Recall@K.

| Model | Amazon | | Yelp | | T-Finance | |
|---|---|---|---|---|---|---|
| Model | w/o DS | w/ DS | w/o DS | w/ DS | w/o DS | w/ DS |
| GCN | 90.18$_{\pm1.01}$ | 65.37$_{\pm0.69}$ | 51.95$_{\pm0.24}$ | 44.04$_{\pm0.27}$ | 89.57$_{\pm0.22}$ | 57.50$_{\pm0.89}$ |
| GAT | 90.22$_{\pm2.18}$ | 66.99$_{\pm0.83}$ | 56.42$_{\pm0.00}$ | 48.11$_{\pm0.00}$ | 82.59$_{\pm0.00}$ | 51.88$_{\pm0.00}$ |
| GraphSAGE | 88.18$_{\pm0.64}$ | 65.20$_{\pm0.61}$ | **58.92**$_{\pm0.23}$ | 51.01$_{\pm0.49}$ | 65.88$_{\pm0.72}$ | 43.68$_{\pm1.22}$ |
| BernNet | 89.74$_{\pm1.65}$ | 65.53$_{\pm1.84}$ | 52.33$_{\pm0.21}$ | 47.89$_{\pm0.25}$ | 86.59$_{\pm0.77}$ | 62.31$_{\pm0.18}$ |
| PCGNN | 89.55$_{\pm0.64}$ | 66.67$_{\pm0.46}$ | 50.89$_{\pm0.12}$ | 43.14$_{\pm0.08}$ | 80.08$_{\pm1.44}$ | 53.36$_{\pm0.69}$ |
| GHRN | 89.55$_{\pm0.00}$ | 62.93$_{\pm0.00}$ | 54.96$_{\pm0.54}$ | 51.13$_{\pm0.57}$ | 87.53$_{\pm0.69}$ | 60.16$_{\pm0.28}$ |
| BWGNN | 90.00$_{\pm0.00}$ | 63.90$_{\pm0.00}$ | 54.98$_{\pm0.83}$ | 50.33$_{\pm0.37}$ | 87.76$_{\pm0.84}$ | 60.75$_{\pm0.72}$ |
| BAT | 90.18$_{\pm1.21}$ | 66.99$_{\pm0.83}$ | 56.80$_{\pm2.64}$ | 51.37$_{\pm2.14}$ | 88.83$_{\pm1.54}$ | 56.54$_{\pm0.72}$ |
| V-REx | 89.24$_{\pm0.21}$ | 62.76$_{\pm0.92}$ | 54.94$_{\pm0.25}$ | 51.37$_{\pm0.54}$ | 86.82$_{\pm0.19}$ | 59.72$_{\pm0.28}$ |
| GroupDRO | 89.55$_{\pm0.00}$ | 61.63$_{\pm0.61}$ | 55.63$_{\pm0.54}$ | 51.09$_{\pm0.40}$ | 88.63$_{\pm0.29}$ | 60.53$_{\pm0.31}$ |
| SRGNN | 90.00$_{\pm0.37}$ | 63.41$_{\pm0.40}$ | 54.31$_{\pm0.58}$ | 50.79$_{\pm0.59}$ | 87.92$_{\pm0.73}$ | 59.20$_{\pm1.10}$ |
| GDN | 89.39$_{\pm0.21}$ | 64.88$_{\pm1.05}$ | 49.34$_{\pm1.66}$ | 48.21$_{\pm1.64}$ | 85.73$_{\pm0.22}$ | 58.02$_{\pm0.91}$ |
| **HEM(+GCN)** | **90.45**$_{\pm0.00}$ | 65.85$_{\pm0.00}$ | 52.83$_{\pm0.49}$ | 49.47$_{\pm0.20}$ | 90.27$_{\pm0.44}$ | 62.23$_{\pm0.58}$ |
| **HEM(+BWGNN)** | 90.15$_{\pm0.21}$ | **67.80**$_{\pm0.80}$ | 56.10$_{\pm0.68}$ | **52.63**$_{\pm0.28}$ | **90.43**$_{\pm0.22}$ | **63.12**$_{\pm0.10}$ |

into test data('w/ DS'). Furthermore, we divide the 'w/o DS' domain into training, validation, and test('w/o DS') data. Since graph anomaly detection datasets are usually with severe data imbalance, we use AUPRC (Area Under the Precision-Recall Curve) and Recall@K (the recall among the top-K highest-confidence predictions) as the evaluation metrics.

## 4.4 Results

Based on the results in Table 1, we can get the following observations:

- Baselines all fail significantly under structural distribution shift: (1) Although baseline methods performs well on test data without distribution shift, they all experience a significant performance drop on test data with structural distribution shift. This phenomenon indicates that these methods may rely on spurious patterns that can not be generalized to the test ('w/ DS') environment.

Table 3: Ablation studies on ego-neighborhood disentangled encoder and homophily-aware environment mixup, where 'w/o END' denotes removing the ego-neighborhood disentangled encoder, 'w/o DIV' denotes removing diversity loss and 'w/o PER' denotes removing PER loss. The best results are in bold and the second-best results are underlined. The results are reported in AUPRC.

| Dataset | Amazon | | Yelp | | T-Finance | |
| Model | w/o DS | w/ DS | w/o DS | w/ DS | w/o DS | w/ DS |
|---|---|---|---|---|---|---|
| w/o END encoder | $91.41_{\pm 0.43}$ | $66.59_{\pm 1.63}$ | $58.33_{\pm 1.09}$ | $56.53_{\pm 0.90}$ | $94.52_{\pm 0.23}$ | $62.50_{\pm 0.50}$ |
| w/o DIV | $\mathbf{93.34}_{\pm \mathbf{0.64}}$ | $70.96_{\pm 1.64}$ | $\underline{61.08}_{\pm 0.89}$ | $\underline{57.52}_{\pm 1.18}$ | $94.08_{\pm 0.71}$ | $64.73_{\pm 1.37}$ |
| w/o PER | $\underline{92.45}_{\pm 0.04}$ | $\underline{72.06}_{\pm 0.16}$ | $60.07_{\pm 0.88}$ | $56.00_{\pm 0.55}$ | $\mathbf{94.88}_{\pm \mathbf{0.15}}$ | $65.50_{\pm 0.51}$ |
| **HEM(+BWGNN)** | $91.97_{\pm 0.74}$ | $\mathbf{74.91}_{\pm \mathbf{0.49}}$ | $\mathbf{61.42}_{\pm \mathbf{0.56}}$ | $\mathbf{58.45}_{\pm \mathbf{1.01}}$ | $\underline{94.76}_{\pm 0.06}$ | $\mathbf{66.04}_{\pm \mathbf{0.29}}$ |

For example, a strong baseline GHRN, experiences performance drop of nearly 25%, 8%, 33%. (2) Out-of-distribution baselines all fail to achieve consistent improvement across all datasets. Compared with BWGNN, the backbone classifier used in these OOD methods, some of the OOD baselines can achieve improvement on specific datasets, like EERM in Amazon and V-REx in Yelp. However, none of them achieves consistent generalization improvement across all datasets, indicating these OOD methods rely on diverse training environments with ground-truth environment labels, e.g. V-REx, GroupDRO, SRGNN. EERM employs the REINFORCE algorithm to create diverse training environments by directly optimizing multiple edge masks, which costs too much memory resources.

- Our proposed method achieves consistent performance improvement across all datasets. Compared with the best baselines, our method achieves relative improvements of 3.4%, 6.9%, and 1.3% on test ('w/ DS') data, and outperforms all general and graph-specialized OOD baselines. This implies that our method can capture the invariant patterns consistently, and create new environments in a memory-efficient manner.

## 4.5  Abaltion Study

In this section, we conduct ablation studies to verify the effectiveness of our proposed ego-neighborhood disentangled encoder and homophily-aware environment mixup.

**Ego-neighborhood disentangled encoder**   We remove the disentangled encoder and simply apply a GNN encoder. From Table 3, we can find that without ego-neighborhood disentangled encoder, the model's performance drops significantly in 'w/o DS' and 'w/ DS' test data across all datasets, which implies the disentanglement design is important in modeling feature and structural information in GAD tasks.

**Homophily-aware environment mixup**   We remove the diversity loss ('w/o DIV') and preserved edge ratio loss ('w/o PER') separately, and both of them show significant performance drops in Table 3. Specifically, removing diversity loss leads to consistent performance drop across all datasets, but removing PER loss increases the performance on T-Finance, implying that deleting a large proportion of edges in T-Finance can increase the generalization capability while preserving the performance in the training stage. This can be verified in the hyperparameter sensitivity analysis. Moreover, our proposed method reduced overfitting in the training stage by adding the diversity loss and preserved edge ratio loss. For example, in the Amazon dataset, the models removing diversity loss or removing PER loss perform better in the test('w/o DS') but worse in the test('w/ DS').

## 4.6  Complexity Analysis

In this section, we analyze the computation complexity of **HEM**. Denote $|V|$ and $|E|$ as the total number of nodes and edges in the graph, and denote $d$ as the dimension of hidden representation. The ego-neighborhood disentangled encoder has a time complexity of $\mathcal{O}(|V|d^2 + |E|d)$. The homophily-aware environment mixup has a time complexity of $\mathcal{O}(|E|d + |E|d + |E|) = \mathcal{O}(|E|d)$, including the edge reweighting and computation of diversity loss and PER loss. Therefore, the total computation complexity for **HEM** is $\mathcal{O}(|V|d^2 + |E|d)$. **HEM** has a linear computation complexity with respect to the number of nodes and edges, comparable to existing GNN baselines.

# 5    Related Work

In this section, we review the existing works about graph anomaly detection and out-of-distribution generalization.

**Graph Anomaly Detection**    The common challenges in graph anomaly detection are data imbalance and heterophily[1], and extensive works are proposed to solve these problems from spatial and spectral prospectives[30–34]. For spatial methods, PCGNN[28] proposes a balanced sampler for the neighborhood aggregation process in GNNs to reduce data imbalance. BAT[29] further proposes a class-rebalancing-free data augmentation framework based on a topological paradigm, which can mitigate the class-imbalance bias and achieve consistent performance boosting across general class imbalance node classification tasks. GAS[35] utilizes structural preprocessing to handle severe heterophily before applying the GNN model. CARE-GNN[36], PMP[37], GraphConsis[38] propose to design new message passing and aggregation mechanisms to mitigate the influence of heterophily on prediction. Another type of method like H2-FDetector[39] decouples the information aggregation for homophilic and heterophilic patterns. Since traditional graph convolution is known as low-frequency filters[40], extensive spectral GNN methods are proposed to handle high-frequency signals in heterophilic graphs[41][42]. BWGNN[18] propose a polynomial spectral GNN with Beta kernel as prior, which is proved to fit the spectral energy distribution in GAD scenario and achieves obvious performance improvement compared with BernNet[26], which has no prior and tries to learn a kernel from all possible polynomial functions. AMNet[43] tries to learn high-frequency and low-frequency signals separately and combines them for prediction. While these GAD-specific methods have achieved substantial progress, there is still room for further enhancement. They fail to adequately model the interaction patterns between the ego-node and its surrounding neighborhood, which could potentially reveal discrepancies that are crucial for more accurate anomaly identification.

**Out-of-Distribution in Graph (OOD)**    The structural distribution shift is a prevalent issue in graph data. For example, variations in disciplines and time periods may result in different citation patterns in citation networks, and variations in geographic locations and relationship types result in different interaction types in social networks. Existing OOD methods can be divided into general methods and graph specialized methods[44–51]. General OOD approaches, such as V-REx[20] and GroupDRO[21], commonly depend on diverse training environments characterized by explicit environment labels, a challenging requirement in graph anomaly detection where inherent homophily obscures environment distinctions. While graph-specific OOD methods have emerged, limitations persist. For instance, EERM[14] constructs novel training environments using learnable graph editors and trains with V-REx to learn invariant representations, yet incurs significant memory overhead due to maintaining multiple full adjacency matrices. SRGNN[22] utilizes the kernel mean matching method to regularize the discrepancy between embeddings of training and test data, while the lack of supervised information leads to its limited performance. GDN[12, 8] is the first work to address structural distribution shift in graph anomaly detection. They handle the problem in an invariant learning way by identifying the critical anomaly features with gradients, while neglecting the importance of varying local structures in distribution shift. In conclusion, prevailing methodologies still have limitations prominently including memory inefficiency, insufficient availability of supervised information, neglect of inherent local structural patterns, and undue reliance on specific training environment assumptions.

# 6    Conclusion

In this paper, we proposed the Ego-Neighborhood Disentangled Encoder with **H**omophily-aware **E**nvironment **M**ixup (**HEM**), for improving the OOD generalization performance for GAD problems. First, we proposed an ego-neighborhood disentangled encoder to disentangle feature and structural representations, in order to model discrepancy patterns and discover invariant patterns. Then, we proposed the homophily-aware environment mixup to generate training environments with diverse local structures to improve the generalization capability of the model. Our model achieves consistent improvement across all datasets.

## Acknowledgments and Disclosure of Funding

This work was supported by the National Key Research and Development Program of China (No. 2022ZD0115903, 2023YFF1205001), National Natural Science Foundation of China No. 62222209, Beijing National Research Center for Information Science and Technology under Grant No. BNR2023TD03006, and Beijing Key Lab of Networked Multimedia.

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

# A   Datasets

## A.1   Dataset Details

- **Amazon**. The task of the Amazon dataset is to identify anomalous users who wrote fake reviews for products under the Musical Instrument category on the Amazon website. Uses are nodes in the graph and there are three types of edges: U-P-U (connects users reviewing at least one same product), U-S-U (connects users giving at least one same star rating within one week), U-V-U (connects users with top 5% mutual review text similarities among all users). The node features are handcrafted features about user information.

- **Yelp**. Yelp dataset includes hotel and restaurant reviews, and the task is to filter spam reviews. The reviews are nodes in the graph and there are three types of edges: R-U-R (connects reviews posted by the same person), R-S-R (connects reviews about the same product with the same rating), R-T-R (connects reviews about the same product while posted in the same month). The node features are handcrafted features from raw texts.

- **T-Finance**. T-Finance dataset is about a transaction network and the target is to find anomaly accounts. The nodes are accounts and the edges are transactions between accounts. The node features are user profile details like registration days.

## A.2   Dataset Split Method

The dataset is divided by the homophily ratio of nodes, split into 50%/10%/20%/20% for training, validation, in-distribution test, and out-of-distribution (OOD) test sets, respectively.

The bottom 20% of nodes with the lowest homophily ratios are selected to form the OOD test set.

# B   Baseline Models

- **General GNNs**: GCN[24] utilizes convolution operation to aggregate messages from the neighborhood and achieves excellent performance in semi-supervised graph tasks. GAT[23] adds an attention mechanism to the message passing process, assigning different weights to edges in order to aggregate information from important neighbors. GraphSAGE[25] utilizes neighborhood sampling methods to fit inductive learning.

- **GNNs Specialized for Graph Anomaly Detection**: BernNet[26] utilizes a K-order Bernstein polynomial approximation to estimate arbitrary graph spectral filers. BWGNN[18] uses Beta distribution as prior for spectrum distribution of graph anomaly detection data. GHRN[27] drops inter-class edges to delineate high-frequency components and increases the homophily of heterophilic dataset. PCGNN[28] designs a label-balanced sampler to sample nodes and edges within neighborhoods to increase homophily.BAT[29] proposed a plug-and-play framework that dynamically augments the graph topology via uncertainty-based node risk estimation and virtual connections.

- **General Out-of-Distribution Methods**: GroupDRO[21] puts more weights on worst-case across different training domains. V-REx[20] enhances out-of-distribution generalization by extrapolating risks and minimizing the variance of losses across different environments.

- **Graph Out-of-Distribution Methods**: SRGNN[22] uses kernel mean matching to regularize the difference in the distributions of training and test nodes' embeddings, in order to avoid learning spurious embedding distribution. EERM[14] utilizes REINFORCE to create diverse environments at the training stage to enhance generalization capability. GDN[12] designs invariance losses and uses gradient descent on input features to filter invariant features.

# C   Hyperparameter Sensitivity Analysis

## C.1   Edge Preservation Ratio

We analyze the sensitivity of the hyperparameter $\rho$, the edge preservation ratio in PER loss from Equation 13. From Figure 2, we can find a proper $\rho$ is neither too small nor too big. A small $\rho$ means preserving only a small proportion of edges, which may destroy the invariant patterns, and a big $\rho$ lead to a structural distribution without much shift. A perfect choice of $\rho$ is a balance of both aspects.

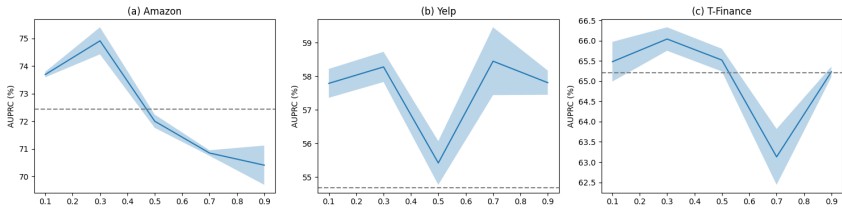

Figure 2: Sensitivity of hyperparameter $\rho$. The area shows the average AUPRC and standard deviations in the test('w/ DS') data. The dashed line represents the average AUPRC of the best-performed baseline.

## C.2 Advaserarial Training

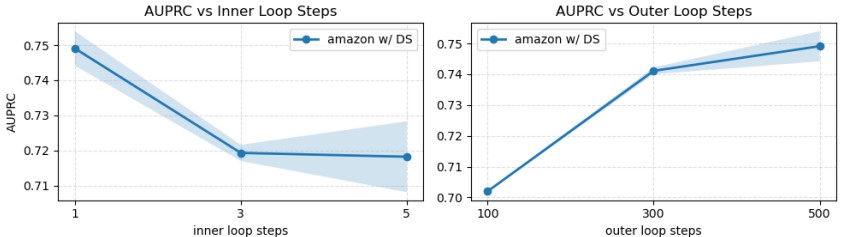

Figure 3: Sensitivity of inner/outer training steps in the adversarial training. The area shows the average AUPRC and standard deviations in the amazon test('w/DS'). By default we choose inner loop steps = 1 and outer loop steps = 500.

We conducted an extensive analysis on the Amazon dataset to evaluate the sensitivity of HEM to the number of inner loop and outer loop steps. Our findings indicate that for the outer loop steps, the model demonstrates robust performance as long as the training steps are not excessively limited. Conversely, the model exhibits relatively higher sensitivity to the inner loop steps. Utilizing a smaller number of inner loop training steps allows for the generation of augmented graphs with an appropriate level of difficulty, thereby facilitating easier model convergence.

## D Case Study

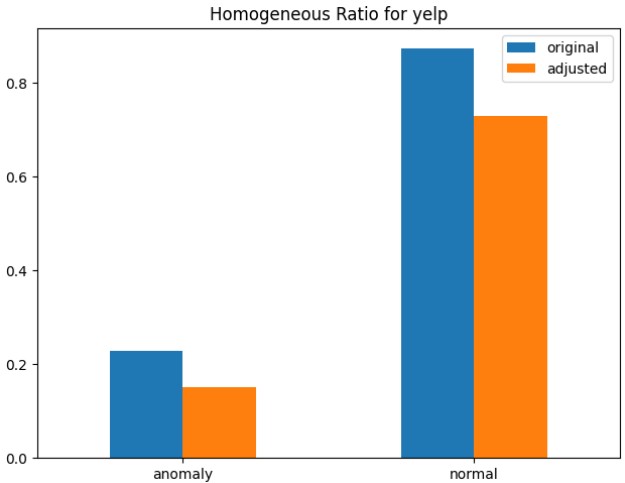

Figure 4: Case Study of Homophily Ratio in Yelp Dataset.

In the case study, we analyze the homophily ratio of anomaly and normal nodes in Yelp dataset. During the data preprocessing, we construct structural distribution shift by sampling nodes with high homophily ratio as training data, leading to a shortcut to predict the label of central node by its neighbors. From Figure 4, the model has learnt to adjust edge weight to decrease the homophily ratio of the total graph and discovered patterns based on the augmented graph.

# E Theoretical Proofs

## E.1 Lemma 1 (Predictor Dependence on Homophily)

The optimal coefficient that minimizes the Mean Squared Error (MSE) between predictions and labels is

$$\alpha^*(h) = \frac{\text{Cov}(Y, r)}{\text{Var}(r)} = \frac{\pi(1 - \pi)(2h - 1)}{\frac{h(1-h)}{d} + \pi(1 - \pi)(2h - 1)^2}. \tag{20}$$

Firstly, the value of some key statistical properties are:

- $\text{Cov}(Y, r) = \pi(1 - \pi)(2h - 1)$,
- $\text{Var}(Y) = \pi(1 - \pi)$,
- $\text{Var}(r \mid Y=y) = \frac{h(1-h)}{d}$,
- $\text{Var}(r) = \frac{h(1-h)}{d} + \pi(1 - \pi)(2h - 1)^2$.

For a one-layer message-passing GNN followed by a linear classifier, the node embedding $Z_v$ is:

$$Z_v = (AX)_v = r_v \mu_1 + (1 - r_v)\mu_0 = \mu_0 + r_v \Delta\mu. \tag{21}$$

The final prediction $\hat{Y}_v$ is then:

$$\hat{Y}_v = w^\top \mu_0 + \underbrace{(w^\top \Delta\mu)}_{\alpha} r_v. \tag{22}$$

The optimal coefficient $\alpha^\star$ that minimizes the Mean Squared Error (MSE) is given by linear regression:

$$\alpha^\star = \frac{\text{Cov}(Y, r)}{\text{Var}(r)} = \frac{\pi(1 - \pi)(2h - 1)}{\frac{h(1-h)}{d} + \pi(1 - \pi)(2h - 1)^2}. \tag{23}$$

**Lemma 2 (Bounded Deviation on the Test Distribution).** If there exists an $\epsilon$ such that $|h_{\text{te}} - h_{\text{tr}}| \leq \epsilon$, with the proposed environment mixup and adversarial training, the gap between the expected test loss of the training-optimal parameters and the optimal test loss is bounded. Specifically,

$$L(h_{\text{te}}, \alpha^*_{\text{tr}}) - L(h_{\text{te}}, \alpha^*_{\text{te}}) \leq V_{\max} L_f \epsilon^2 \tag{24}$$

, where $L_f$ denotes the Lipschitz constant of $f(h) = \alpha^*_h$, and $V_{\max}$ represents the maximum of $Var(r)$ for $h$ satisfying $|h - h_{\text{tr}}| \leq \epsilon$.

For simplicity, we define:

- $C(h) = \text{Cov}(Y, r) = \pi(1 - \pi)(2h - 1)$,
- $V(h) = \text{Var}(r) = \frac{h(1-h)}{d} + \pi(1 - \pi)(2h - 1)^2$.

Then the MSE Loss is:

$$L(h; \alpha) = \mathbb{E}\big[(Y - \hat{Y})^2 \mid h\big] = V(h) - 2\alpha C(h) + \alpha^2 V(h). \tag{25}$$

Define the optimal $\alpha$ on $h$ as $\alpha^\star = f(h) = \frac{C(h)}{V(h)}$, and the optimal loss as:

$$L^\star = V(h) - \frac{C(h)^2}{V(h)}. \tag{26}$$

We assume HEM trains on $\mathcal{H}_\epsilon = [h_{\mathrm{tr}} - \epsilon,\ h_{\mathrm{tr}} + \epsilon]$ by applying environment mixup, which covers $h_{\mathrm{te}}$.

Define:

- $f_{\min} = \inf_{h \in \mathcal{H}_\epsilon} f(h)$,
- $f_{\max} = \sup_{h \in \mathcal{H}_\epsilon} f(h)$,
- $V_{\max} = \sup_{h \in \mathcal{H}_\epsilon} V(h)$.

Then

$$L(h; \alpha) = L^\star(h) + V(h)\big(\alpha - f(h)\big)^2, \tag{27}$$

therefore we can get the upper bound of the loss difference:

$$L(h; \alpha) - L^\star(h) \le \sup_{h \in \mathcal{H}_\epsilon} V(h)\big(\alpha - f(h)\big)^2 \le V_{\max}\left(\sup_{h \in \mathcal{H}_\epsilon}\big(\alpha - f(h)\big)^2\right).$$

Since the goal of adversarial training in HEM is to minimize the loss in the worst case, we further assume HEM can minimize the upper bound, then:

$$\alpha = \frac{f_{\min} + f_{\max}}{2}. \tag{28}$$

Thus,

$$L(h; \alpha) - L^\star(h) \le V_{\max}\left(\sup_{h \in \mathcal{H}_\epsilon}\left(\frac{f_{\max} - f_{\min}}{2}\right)^2\right). \tag{29}$$

If $f$ follows Lipschitz continuity, then

$$f_{\max} - f_{\min} \le L_f\big((h_{\mathrm{tr}} + \epsilon) - (h_{\mathrm{tr}} - \epsilon)\big) = 2L_f\epsilon. \tag{30}$$

Finally,

$$L(h; \alpha) - L^\star(h) \le V_{\max} L_f^2 \epsilon^2. \tag{31}$$

# F   Configurations

All experiments are conducted with:

- Operating System: Ubuntu 20.04.6 LTS
- CPU: Intel(R) Xeon(R) Gold 6348 CPU @ 2.60GHz
- GPU: NVIDIA GeForce RTX 4090 with 24 GB of memory
- Software: Python 3.12.4; CUDA 12.2; PyTorch 2.4.1

