# OpenReview forum: "Out-of-Distribution Generalized Graph Anomaly Detection with Homophily-aware Environment Mixup"
_NeurIPS.cc/2025/Conference — NeurIPS 2025 poster_

### Official Review · Reviewer_i6um · 2025-06-04

**Clarity:** 2
**Significance:** 2
**Originality:** 2
**Rating:** 3
**Confidence:** 4

**Summary:**

The paper addresses the challenge of graph anomaly detection under structural distribution shifts due to selection bias. It proposes a novel method called HEM (Ego-Neighborhood Disentangled Encoder with Homophily-aware Environment Mixup) to discover invariant patterns and dynamically adjust edge weights to handle varying homophilic structures. The approach involves disentangling feature and structure embeddings, and performing adversarial training to improve generalization capabilities.

**Questions:**

1. How does the disentangled encoder theoretically ensure prediction stability by separating features and structures?
2. Is there a unified mathematical representation of "stability" across distributions for invariant patterns, and how is it proven?
3. Could you elaborate on the specific criteria or methods used to distinguish and partition the datasets into these two domains?
4. The method seems to primarily address a unidirectional distribution shift. How does it ensure the model’s stability in scenarios where homophily either intensifies or remains nearly constant?

**Ethical Concerns:**

["NO or VERY MINOR ethics concerns only"]

**Final Justification:**

I appreciate the thoughtful rebuttal addressing most of my concerns. I decided to maintain my original score.

**Limitations:**

1. The baselines are outdated, as they do not include recent OOD detection methods (like energy-based models, uncertainty-aware models) or some recent anomaly detection frameworks. Notably, there have already been frameworks that decouple attribute and structure, such as GOOD-D proposed in WSDM23.
2. As a typical binary classification problem, using only AUPRC as the metric is insufficient to convince readers.
3. The paper does not elaborate on how General GNNs (e.g., GCN, GAT) are adapted for anomaly detection tasks.

**Quality:**

2

**Strengths And Weaknesses:**

Strength:
1. Innovative decoupling framework.
2. Dynamic & continuous environment augmentation for distribution shifts.
Weakness:
1. Outdated baselines, missing recent frameworks.
2. Lacks mathematical proofs for invariant pattern stability.
3. Overly complex descriptions.

---

> ### Author Rebuttal · Authors · 2025-07-30
>
> ## Q1&Q2
>
> **Annotations**
>
> Let $h$ denote the homogeneous ratio and $Y$ be the node label, where $Y=1$ for anomalous nodes and $Y=0$ for normal nodes. The proportion of anomalous nodes is $\pi$, such that $P(Y=1) = \pi$.
>
> The proportion of anomalous neighbors for a node $u$, denoted as $r_u$, is defined as:
> $$r_u = \frac{1}{\text{deg}(u)}\sum_{(u,v)\in E} Y_v$$
> We assume a constant node degree $d$, $\mu_1=\mathbb E[X\mid Y{=}1]$ and $\mu_0=\mathbb E[X\mid Y{=}0]$, with the assumption that $\Delta\mu=\mu_1-\mu_0\neq 0$.
>
> **Theorem 1:** Under a homophily distribution shift, the optimal parameter $w^\star$ learned on a training set with homophily $h_{tr}$ is no longer optimal for a test set with a different homophily $h_{te}$.
>
> Key statistical properties:
>
> * $\operatorname{Cov}(Y,r)=\pi(1-\pi)(2h-1)$
> * $\operatorname{Var}(Y)=\pi(1-\pi)$
> * $\operatorname{Var}(r\mid Y{=}y)=\frac{h(1-h)}{d}$
> * $\operatorname{Var}(r)=\frac{h(1-h)}{d}+\pi(1-\pi)(2h-1)^2$
>
> For a one-layer message-passing GNN followed by a linear classifier, the node embedding $Z_v$ is:
> $Z_v=(AX)_v=r_v\mu_1+(1-r_v)\mu_0=\mu_0+r_v\Delta\mu$
>
> The final prediction $\hat{Y}_v$ is then: $\hat{Y}_v$
>
> $=w^\top\mu_0+\underbrace{(w^\top\Delta\mu)}_{\alpha}\,r_v$
>
> The optimal coefficient $\alpha^{\star}$ that minimizes the Mean Squared Error (MSE) is given by linear regression:$$\alpha^{\star}=\frac{\operatorname{Cov}(Y,r)}{\operatorname{Var}(r)} =\frac{\pi(1-\pi)(2h-1)}{\frac{h(1-h)}{d}+\pi(1-\pi)(2h-1)^2}$$
> Since $\alpha^{\star}$ is a function of the homophily ratio $h$, the optimal weight vector $w^\star$ also depends on $h$. Consequently, a parameter $w^\star$ learned with $h_{tr}$ will be suboptimal for a test distribution with $h_{te} \neq h_{tr}$, causing model **instability**.
>
> **Theorem 2**: If there exists an $\epsilon$ such that $|h_{te}-h_{tr}|\leq \epsilon$, then the difference between the empirical loss and the optimal loss on the test set is bounded. Specifically, $L(h_{te};\alpha) - L^\star(h_{te}) \leq V_{max}L_f\epsilon^2$, where $L_f$ is the Lipschitz constant of $f$.
>
> Define
> $C(h)=\operatorname{Cov}(Y,r)=\pi(1-\pi)\,(2h-1)$
> $V(h)=\operatorname{Var}(r)=\frac{h(1-h)}{d}+\pi(1-\pi)(2h-1)^2$
>
> The MSE loss is
> $L(h;\alpha)=E[(Y-\hat{Y})^2|h]=V(h)-2\alpha C(h)+\alpha^2 V(h)$
>
> The optimal $\alpha$ for a given $h$ is defined as $\alpha^{\star}=f(h)=\frac{C(h)}{V(h)}$, with corresponding optimal loss $L^\star=V(h)-\frac{C(h)^2}{V(h)} $.
>
> We assume that HEM trains on the interval ${H}_\epsilon =$
>
> $[h_{tr}-\epsilon, h_{tr}+\epsilon]$,
>
> which encompasses $h_{te}$.
>
> Define
>
> * $f_{min}=\inf_{h\in \mathcal{H}_\epsilon} f(h)$
> * $f_{max}=\sup_{h\in \mathcal{H}_\epsilon} f(h)$
> * $V_{max}=\sup_{h\in \mathcal{H}_\epsilon} V(h)$
>
>
> $L(h;\alpha)=L^\star(h) + V(h)(\alpha-f(h))^2$
>
> Therefore, we can establish an upper bound for the empirical loss exceeding the optimal loss:
>
> $L(h;\alpha) - L^\star(h) \leq \sup_{h\in \mathcal{H}_\epsilon} V(h)(\alpha-f(h))^2$
>
> $\leq V_{max}(\sup_{h\in \mathcal{H}_\epsilon}(\alpha-f(h))^2)$
>
> Since the objective of the adversarial training process in HEM is to minimize the loss in the worst case, we further assume HEM minimizes this upper bound. This implies that $\alpha$ is chosen as the midpoint of the range of $f(h)$:
>
> $\alpha=\frac{f_{min}+f_{max}}{2}$
>
> $L(h;\alpha) - L^\star(h) \leq V_{max}(\sup_{h\in \mathcal{H}_\epsilon} (\frac{fmax- fmin}{2})^2) )$
>
> If $f$ adheres to Lipschitz continuity,
>
> $f_{max}-f_{min}\leq L_f((h_{tr}+\epsilon)-(h_{tr}-\epsilon))=2L_f\epsilon$
>
> Finally,
> $L(h;\alpha) - L^\star(h) \leq V_{max}\left(\frac{2L_f\epsilon}{2}\right)^2 = V_{max}(L_f\epsilon)^2 = V_{max}L_f^2\epsilon^2$
>
> Theorem 2 shows that HEM ensures stability through adversarial training on disentangled embeddings, resulting in a bounded test loss, unlike traditional methods, which lack this condition and thus may suffer unbounded loss.
>
> ## Q3
>
> The dataset is divided by the *homogeneous ratio* of nodes,  split into 50%/10%/20%/20% for training, validation, in-distribution test, and out-of-distribution (OOD) test sets, respectively. The bottom 20% of nodes with the lowest homogeneous ratios are selected to form the OOD test set.
>
> ## Q4
>
> To ensure that the model remains stable under minimal distributional shift, we report its performance on the test sets without/with distribution shift (w/o DS and w/ DS). As shown, our model achieves state-of-the-art or near-SOTA performance even on the w/o DS test set.
>
> ## L1
> (1) We add a new baseline [ICML 2024] BAT [1], an uncertainty-aware framework.
> - |      | ('amazon', 'w/o DS') | ('amazon', 'w/ DS') | ('yelp', 'w/o DS') | ('yelp', 'w/ DS')  | ('tfinance', 'w/o DS') | ('tfinance', 'w/ DS') |
> | :--- | :------------------- | :------------------ | :----------------- | :----------------- | :--------------------- | :-------------------- |
> | BAT  | **0.9403(0.0204)**   | 0.7226(0.0116)      | **0.6219(0.0261)** | 0.5493(0.0049)     | 0.9277(0.0067)         | 0.5635(0.0175)        |
> | HEM  | 0.9197(0.0074)       | **0.7491(0.0049)**  | 0.6142(0.0056)     | **0.5845(0.0101)** | **0.9476(0.0060)**     | **0.6604(0.0029)**    |
>
> (2) Key Differences between HEM and GOOD-D:
> 1. Problem Setting:
>
> - **GOOD-D** focuses on **unsupervised graph-level out-of-distribution (OOD) detection**, where the goal is to distinguish whether an entire graph is from the in-distribution or OOD.
> - **HEM** is designed for **node-level graph anomaly detection** under **structural distribution shift**, specifically addressing the performance degradation caused by varying levels of homophily between training and test data.
>
> 2. Disentanglement Strategy:
>
> - In **GOOD-D**, disentanglement is implemented through a **perturbation-free augmentation** strategy that generates a structure view using pre-computed structural encodings, contrasting it with a feature view.
> - In **HEM**, we propose an **ego-neighborhood disentangled encoder** that **explicitly separates the modeling of node features and neighborhood structures through MLP and GNN sharing the same encoder**, allowing more targeted learning of invariant patterns for node-level prediction.
>
> 3. **Optimization Objectives and Mechanisms:**
>
> - **GOOD-D** uses a **hierarchical contrastive learning framework** to capture ID patterns and define OOD scores based on multi-level inconsistencies.
> - **HEM** introduces a **homophily-aware environment mixup** that **dynamically adjusts edge weights** to create structurally diverse environments, and optimizes the encoder in an **adversarial training** setup to improve robustness across distribution shifts.
>
> ## L2
> To address this concern, we have supplemented our evaluation with Rec@K metric. Since GAD is a highly imbalanced classification problem, we believe AUPRC and Rec@K are more informative than AUROC and ACC.
>
> - **Rec@K (Recall at K)** measures the recall among the top-K highest-confidence predictions, where K equals the number of actual anomalies in the test set.
>
> |           | ('amazon', 'w/o DS') | ('amazon', 'w/ DS') | ('yelp', 'w/o DS') | ('yelp', 'w/ DS')  | ('tfinance', 'w/o DS') | ('tfinance', 'w/ DS') |
> | :-------- | :------------------- | :------------------ | :----------------- | :----------------- | :--------------------- | :-------------------- |
> | GCN       | 0.9018(0.0101)       | 0.6537(0.0069)      | 0.5195(0.0024)     | 0.4404(0.0027)     | 0.8957(0.0022)         | 0.575(0.0089)         |
> | GAT       | **0.9022(0.0218)**   | 0.6699(0.0083)      | 0.5642(0.0)        | 0.4811(0.0)        | 0.8259(0.0)            | 0.5188(0.0)           |
> | GraphSAGE | 0.8818(0.0064)       | 0.652(0.0061)       | **0.5892(0.0023)** | 0.5101(0.0049)     | 0.8867(0.0169)         | 0.5329(0.0258)        |
> | BernNet   | 0.8974(0.0165)       | 0.6553(0.0184)      | 0.5233(0.0021)     | 0.4789(0.0025)     | 0.8659(0.0077)         | 0.6231(0.0018)        |
> | PCGNN     | 0.8955(0.0064)       | 0.6667(0.0046)      | 0.5089(0.0012)     | 0.4314(0.0008)     | 0.8008(0.0144)         | 0.5336(0.0069)        |
> | GHRN      | 0.8955(0.0)          | 0.6293(0.0)         | 0.5496(0.0054)     | 0.5113(0.0057)     | 0.8753(0.0069)         | 0.6016(0.0028)        |
> | BWGNN     | 0.9(0.0)             | 0.639(0.0)          | 0.5498(0.0083)     | 0.5033(0.0037)     | 0.8776(0.0084)         | 0.6075(0.0072)        |
> | V-REx     | 0.8924(0.0021)       | 0.6276(0.0092)      | 0.5494(0.0025)     | 0.5137(0.0054)     | 0.8682(0.0019)         | 0.5972(0.0028)        |
> | GroupDRO  | 0.8955(0.0)          | 0.6163(0.0061)      | 0.5563(0.0054)     | 0.5109(0.004)      | 0.8863(0.0029)         | 0.6053(0.0031)        |
> | SRGNN     | 0.9(0.0037)          | 0.6341(0.004)       | 0.5431(0.0058)     | 0.5079(0.0059)     | 0.8792(0.0073)         | 0.592(0.011)          |
> | GDN       | 0.8939(0.0021)       | 0.6488(0.0105)      | 0.4934(0.0166)     | 0.4821(0.0164)     | 0.8573(0.0022)         | 0.5802(0.0091)        |
> | BAT       | 0.9018(0.0121)       | 0.6699(0.0083)      | 0.568(0.0264)      | 0.5137(0.0214)     | 0.8883(0.0154)         | 0.5654(0.0072)        |
> | HEM       | 0.9015(0.0021)       | **0.678(0.008)**    | 0.561(0.0068)      | **0.5263(0.0028)** | **0.9043(0.0022)**     | **0.6312(0.001)**     |
>
> ## L3
>
> Although anomaly detection involves significant class imbalance, it can still be formulated as a classification task. Therefore, general GNNs like GCN and GAT can be directly applied.
> Unlike prior works that adopt naive implementations of these GNNs as weak baselines, we follow the insights from [2] by incorporating residual connections and Layer Normalization. These enhancements significantly improve the performance of classic GNNs in GAD tasks.
>
> [1] Liu Z, Qiu R, Zeng Z, et al. Class-imbalanced graph learning without class rebalancing[J]. arXiv preprint arXiv:2308.14181, 2023.
>
> [2] Luo Y, Shi L, Wu X M. Classic gnns are strong baselines: Reassessing gnns for node classification[J]. Advances in Neural Information Processing Systems, 2024, 37: 97650-97669.

---

> > ### Comment · Reviewer_i6um · 2025-08-05
> >
> > I appreciate the thoughtful rebuttal addressing most of my concerns. I decided to raise my score.

---

### Official Review · Reviewer_r6WM · 2025-07-02

**Clarity:** 3
**Significance:** 4
**Originality:** 3
**Rating:** 5
**Confidence:** 4

**Summary:**

This paper addresses the problem of graph anomaly detection (GAD) under structural distribution shifts, where models trained on data with high homophily perform poorly on test data with low homophily. The authors argue that existing methods learn "homophilic shortcuts" that fail to generalize. To tackle this, they propose HEM (Ego-Neighborhood Disentangled Encoder with Homophily-aware Environment Mixup). Experiments on three real-world datasets show that HEM outperforms a range of baselines, especially in the presence of distribution shift.

**Questions:**

1.  The homophily-aware environment mixup learns to re-weight edges to create more challenging training scenarios. How does the framework ensure that the generated edge weights lead to meaningful and realistic low-homophily structures, rather than simply learning a trivial adversarial perturbation that inverts embeddings but does not reflect real-world structural shifts?
2.  The core problem is framed as generalizing from high-homophily training data to low-homophily test data. How would HEM be expected to perform in the reverse scenario (training on low-homophily data and testing on high)? Furthermore, how does it compare to methods designed specifically for static heterophilic graphs, where low homophily is the default and not the result of a distribution shift?

**Ethical Concerns:**

["NO or VERY MINOR ethics concerns only"]

**Final Justification:**

I maintain my review.

**Limitations:**

Yes

**Paper Formatting Concerns:**

No.

**Quality:**

3

**Strengths And Weaknesses:**

pros:
1. the paper provides a well-motivated approach to address this Structural distribution shifts.
2. The proposed HEM framework is novel in its approach to this problem. The combination of a disentangled encoder to separate ego and neighborhood information with a learnable, homophily-aware data augmentation module is a clever and well-integrated solution.
3. The experimental setup is thorough. The authors compare HEM against a comprehensive set of baselines, including general GNNs, specialized GAD methods, and both general and graph-specific out-of-distribution (OOD) methods. The results consistently demonstrate the superiority of HEM on test sets with distribution shifts, validating the effectiveness of the proposed approach.

cons:
1. While generally well-written, the overall framework is quite complex, involving multiple modules and loss functions. The adversarial training dynamic could be explained more intuitively. The term "environment" is used to describe the result of re-weighting edges on a single graph, which might be slightly confusing for readers familiar with OOD literature where environments are often distinct datasets.
2. The paper claims to disentangle feature and structural representations. However, this is achieved structurally by using separate encoders. The paper lacks a quantitative analysis to prove that the representations are truly disentangled (e.g., showing that the neighborhood embedding is more sensitive to structural changes while the ego embedding remains stable). The benefit is only demonstrated through downstream performance.

---

> ### Author Rebuttal · Authors · 2025-07-30
>
> 1. Our method directly confronts this issue by creating augmented environments with diverse homophily distributions. The key insight is that these augmented samples do not need to perfectly mirror realistic data. Their strategic purpose is to create challenging scenarios that **break the spurious connection** between the proportion of anomaly nodes in neighborhood and node labels in high-homophily data.
> By forcing the model to perform well across generated environments—from high to low homophily—we compel it to learn features that are robust and invariant to the graph's homophily level. This process prevents the model from learning a trivial adversarial perturbation (e.g., simply inverting embeddings), as such a simplistic strategy would perform poorly on the original, un-augmented training environments. Ultimately, this approach produces a model that generalizes more effectively to test sets with unknown or different structural properties.
>
> 2. Although our primary motivation was to address the homophily distribution shift observed when moving from high-homophily to low-homophily data, our method does not explicitly minimize the homogeneous ratio of augmented data during the training phase. Instead, our adversarial training framework aims to generate more challenging samples, thereby enhancing the model's generalization ability.

---

> > ### Comment · Reviewer_r6WM · 2025-08-09
> >
> > Thanks for your reply. I will maintain my positive score.

---

### Official Review · Reviewer_YbZZ · 2025-07-02

**Clarity:** 4
**Significance:** 4
**Originality:** 4
**Rating:** 5
**Confidence:** 4

**Summary:**

This paper tackles how Graph Anomaly Detection (GAD) models fail when there are structural distribution shifts between training and test data. It points out that models often learn homophilic shortcuts from biased training data, which hurts their performance on test data with different structures.To fix this, the paper introduces a new framework called HEM (Ego-Neighborhood Disentangled Encoder with Homophile-aware Environment Mixup). HEM's main ideas are:an Ego-Neighborhood Disentangled Encoder that processes a node's own features separately from its neighborhood's structure. A Homophily-aware Environment Mixup module that cleverly re-weights graph edges to create diverse training examples. The whole system is trained adversarially, pushing the encoder to learn stable patterns while the mixup module creates tough, new graph structures. The experiments show that HEM works much better than other methods, especially when the graph structure shifts.

**Questions:**

1 Your mixup module aims to create "diverse" environments, but the objective function focuses on making "hard" examples for the classifier. How do you ensure this creates a true diversity of structures, rather than just pushing all augmented graphs to be low-homophily, for example?

2 You used BWGNN as the neighborhood encoder. How much do your results depend on this specific choice? It would strengthen your claims if you showed that the HEM framework also improves the performance of other backbones like GCN or GAT.

**Ethical Concerns:**

["NO or VERY MINOR ethics concerns only"]

**Final Justification:**

This paper is well-written and highly inspiring. The few concerns were satisfactorily addressed in the rebuttal. I believe it fully meets the required standards. Therefore, I give an "Accept."

**Limitations:**

yes

**Paper Formatting Concerns:**

None.

**Quality:**

4

**Strengths And Weaknesses:**

## pros
Significance: The paper tackles a really important and often overlooked problem in GAD—how to make models generalize when the graph structure changes. This is a big deal for real-world uses like finding fraudsters or bots, where the data is always biased.

Originality: The Homophily-aware Environment Mixup idea is quite innovative. Instead of making multiple copies of the graph, which eats up memory, it just adjusts edge weights. This is a much more efficient and fine-grained way to create the diverse data needed for robust training.

Quality: The technical work is solid. The method makes sense, and the experiments are thorough. You compared HEM against a wide variety of relevant models and showed it consistently comes out on top, particularly in the most challenging scenarios. The ablation studies also clearly prove that each part of your framework is necessary.

Clarity: The paper is well-written and easy to understand. You do a great job explaining the problem and your solution. Figure 1 is especially helpful for getting a quick overview of the whole system.

## cons
No Limitations Section: The paper would be stronger if it discussed its own limitations. For example, the model's performance seems to depend quite a bit on the ρ hyperparameter, and it'd be good to acknowledge the challenge of tuning this in practice.

Homophily-aware Link: The connection between the name "homophily-aware" and the mechanism could be clearer. Your mixup module makes the training examples "harder," but it's not fully explained how this ensures a diversity of homophily levels, rather than just always making homophily lower.

Framework Generalizability: You use a strong, specific GNN (BWGNN) as the backbone. It would be great to see if the HEM framework provides the same benefits when used with other common GNNs like GCN or GAT. This would show that the framework itself is what's providing the boost.

---

> ### Author Rebuttal · Authors · 2025-07-30
>
> 1. While our motivation stemmed from the need to mitigate the homophily distribution shift between high- and low-homophily data, our method does not directly minimize the homogeneous ratio of augmented data during training. Rather, our adversarial training framework is designed to create an environment with harder samples, which implicitly fosters improved model generalization.
>
> 2. We change the neighborhood encoder from BWGNN to GCN, HEM also improves the performance.
>
> |            | ('amazon', 'w/o DS') | ('amazon', 'w/ DS') | ('yelp', 'w/o DS') | ('yelp', 'w/ DS')  | ('tfinance', 'w/o DS') | ('tfinance', 'w/ DS') |
> | :--------- | :------------------- | :------------------ | :----------------- | :----------------- | :--------------------- | :-------------------- |
> | GCN        | **0.9445(0.0097)**   | 0.6990(0.0213)      | 0.5363(0.0094)     | 0.4079(0.0173)     | 0.9366(0.0010)         | 0.6033(0.0041)        |
> | HEM (+GCN) | 0.9370(0.0011)       | **0.7059(0.0009)**  | **0.5509(0.0047)** | **0.5205(0.0100)** | **0.9569(0.0008)**     | **0.6497(0.0011)**    |

---

### Official Review · Reviewer_RDof · 2025-07-03

**Clarity:** 3
**Significance:** 4
**Originality:** 4
**Rating:** 5
**Confidence:** 4

**Summary:**

This paper addresses the problem of graph anomaly detection (GAD) under structural distribution shifts, a scenario where models trained on graphs with high homophily perform poorly on test graphs with lower homophily due to selection bias in real-world data. The authors propose a novel framework, HEM (Ego-Neighborhood Disentangled Encoder with Homophily-aware Environment Mixup), to tackle this challenge. The framework consists of two key contributions:
1.  An **Ego-Neighborhood Disentangled Encoder** that separates the learning of a node's own features from its neighborhood's structural information. This is designed to capture invariant patterns from node features while isolating them from the spurious correlations that arise from shifting structural patterns.
2.  A **Homophily-aware Environment Mixup** module that adversarially generates diverse training environments by re-weighting graph edges. This forces the encoder to learn representations that are robust across different structural distributions in a memory-efficient manner.

Through an adversarial training process, the model learns to generalize better to out-of-distribution data. Extensive experiments on three real-world datasets demonstrate that HEM significantly outperforms a wide range of baselines, achieving state-of-the-art performance for GAD under structural distribution shift.

**Questions:**

1.  The core motivation is to learn "invariant patterns" by disentangling ego and neighborhood representations. Could you provide a more direct, qualitative analysis to support this? For instance, using an embedding visualization technique (like t-SNE) to show that the ego-embeddings ($H_{ego}$) for a given class (anomalous/normal) are clustered more tightly across the "w/o DS" and "w/ DS" domains compared to the neighborhood embeddings ($H_{ne}$)? This would provide strong evidence that the disentanglement is working as intended.
2.  Your environment mixup module is positioned as a memory-efficient alternative to methods like EERM. While EERM resulted in an "Out of Memory" (OOM) error on most datasets, it did run on the Amazon dataset. Could you provide a quantitative comparison of the training time and peak memory usage of HEM versus EERM on the Amazon dataset to concretely substantiate this advantage?
3.  You claim that the ego-neighborhood disentangled encoder is a general framework applicable to various GNNs. Have you performed any experiments by substituting the BWGNN neighborhood encoder with other GNN architectures (e.g., GCN, GAT, or GraphSAGE)? Showing that the HEM framework still provides a significant boost would greatly strengthen this claim of generality.
4.  The adversarial training framework involves an inner and outer loop. Can you comment on the stability and convergence of this training process? How sensitive is the model's final performance to the number of training steps in the inner loop (for the environment mixup) versus the outer loop (for the encoder)?

**Ethical Concerns:**

["NO or VERY MINOR ethics concerns only"]

**Final Justification:**

It has addressed my concerns, and I will keep my score.

**Limitations:**

Yes.

**Paper Formatting Concerns:**

No.

**Quality:**

4

**Strengths And Weaknesses:**

**Strengths:**

* The paper tackles a critical and practical problem in graph machine learning. Structural distribution shift is a common issue in real-world applications like financial fraud and bot detection, and the paper clearly motivates how homophily shortcuts can lead to poor generalization. This work represents a significant step towards creating more robust GAD systems.
* The proposed HEM framework is novel and well-conceived. The combination of a disentangled encoder and an adversarial environment generator is an original approach to this problem. The environment mixup mechanism, which uses continuous edge re-weighting via an attention mechanism, is a particularly clever and efficient alternative to prior work like EERM that required manipulating multiple large adjacency matrices.
* The paper demonstrates high quality in its methodology and evaluation. The proposed method is technically sound, and the components are well-integrated into a coherent adversarial learning framework. The experimental evaluation is comprehensive, comparing HEM against an extensive list of 13 baselines across four relevant categories (General GNNs, GAD-specific GNNs, General OOD, and Graph OOD methods)
* The paper is well-written, clearly structured, and easy to follow. The problem is precisely formulated, and the proposed solution is motivated and explained in detail.

**Weaknesses:**

* While the paper's motivation hinges on discovering "invariant patterns", the analysis does not offer a qualitative look into what these patterns are. The case study in Figure 3 shows that the mixup module successfully alters the graph's homophily ratio, but a deeper analysis (e.g., embedding visualization) to show that the ego-embeddings are indeed more "invariant" than the neighborhood embeddings across different distributions would be highly insightful.

---

> ### Author Rebuttal · Authors · 2025-07-30
>
> 1. Our primary objective with environment mixup is to mitigate the spurious connection between the proportion of anomalous nodes in the neighborhood and the label, a connection often induced by high homophily. The overarching goal of the HEM framework is to achieve invariant ego and neighborhood distributions. Consequently, our methodology aims to simultaneously enhance the performance concerning both ego and neighborhood representations. The overall improvement in performance, while evident in the final results, is also directly substantiated by the ablation study, where the significant performance degradation observed when operating "w/o END encoder" clearly demonstrates the crucial role of disentanglement.
>
> 2. | Model | Peak GPU Memory Usage | Training Time |
> | ----- | --------------------- | ------------- |
> | EERM  | 21.05 GB              | 489.19 s      |
> | HEM   | **0.92 GB**           | **45.67 s**   |
>
> 3. We change the neighborhood encoder from BWGNN to GCN, HEM also improves the performance.
>
> |            | ('amazon', 'w/o DS') | ('amazon', 'w/ DS') | ('yelp', 'w/o DS') | ('yelp', 'w/ DS')  | ('tfinance', 'w/o DS') | ('tfinance', 'w/ DS') |
> | :--------- | :------------------- | :------------------ | :----------------- | :----------------- | :--------------------- | :-------------------- |
> | GCN        | **0.9445(0.0097)**   | 0.6990(0.0213)      | 0.5363(0.0094)     | 0.4079(0.0173)     | 0.9366(0.0010)         | 0.6033(0.0041)        |
> | HEM (+GCN) | 0.9370(0.0011)       | **0.7059(0.0009)**  | **0.5509(0.0047)** | **0.5205(0.0100)** | **0.9569(0.0008)**     | **0.6497(0.0011)**    |
>
> 4. We conducted an extensive analysis on the Amazon dataset to evaluate the sensitivity of HEM to the number of inner loop and outer loop steps. Our findings indicate that for the outer loop steps, the model demonstrates robust performance as long as the training steps are not excessively limited. Conversely, the model exhibits relatively higher sensitivity to the inner loop steps. Utilizing a smaller number of inner loop training steps allows for the generation of augmented graphs with an appropriate level of difficulty, thereby facilitating easier model convergence.
>
> | Inner Loop     | ('amazon', 'w/o DS') | ('amazon', 'w/ DS') |
> | :------------- | :------------------- | :------------------ |
> | 1 (By Default) | 0.9197(0.0074)       | **0.7491(0.0049)**  |
> | 3              | **0.9203(0.0007)**   | 0.7193(0.0023)      |
> | 5              | 0.9193(0.0023)       | 0.7182(0.0101)      |
>
> | Outer Loop       | ('amazon', 'w/o DS') | ('amazon', 'w/ DS') |
> | :--------------- | :------------------- | :------------------ |
> | 500 (By Default) | 0.9197(0.0074)       | **0.7491(0.0049)**  |
> | 300              | **0.9208(0.0002)**   | 0.7411(0.0011)      |
> | 100              | 0.9137(0.0001)       | 0.702(0.0003)       |

---

> > ### Comment · Reviewer_RDof · 2025-08-08
> >
> > Thank you for the response. It has addressed my concerns, and I will keep my score.

---

### Note · Authors · 2025-08-15

Thank you to the reviewers and area chairs for your valuable feedback. We believe our responses have resolved the core concerns.

### **Summary of Core Contributions**

Our work, **HEM**, introduces a novel framework to improve the generalization of Graph Anomaly Detection (GAD) by tackling the distribution shift problem between high- and low-homophily data. Our main contributions are as follows:

1. **Ego-Neighborhood Disentangled encoder:** This encoder effectively separates a node's ego features from its neighborhood structural features, helping the model learn patterns that are invariant to homophily changes.
2. **Homophily-aware Environment Mixup:** We designed an adversarial training framework that dynamically adjusts edge weights to generate training samples with varying levels of homophily. This forces the model to learn more robust features by exposing it to a diverse range of structural environments during training.
3. **Generalizability and effectiveness of our method:** We demonstrated the effectiveness of our approach through ablation studies and state-of-the-art performance on multiple datasets. Furthermore, we showed that the HEM framework is highly flexible and can be applied to various GNN backbones.

### **Responses to Key Questions**

- **Effectiveness of disentanglement:** Our ablation study confirmed the critical role of the disentangled encoder. Removing this module led to a significant performance drop, directly proving that disentanglement is essential for learning invariant patterns.
- **Generalizability of framework:** We conducted additional experiments by replacing the BWGNN neighborhood encoder with GCN. The results showed that the HEM framework still provided a significant performance boost.
- **Stability of adversarial training:** Our detailed sensitivity analysis on the Amazon dataset indicated that while the model is relatively sensitive to the number of inner loop steps, the overall training process is stable and robust within a reasonable range.
- **Updated baselines:** We have incorporated a new baseline, BAT (ICML 2024). In our comparison, HEM achieved better performance than BAT across all datasets.
- **Evaluation metrics:** We supplemented our evaluation with the Rec@K metric to better reflect the challenges of class imbalance in anomaly detection tasks.

---

### Decision · Program_Chairs · 2025-09-17

**Decision:**

Accept (poster)

**Comment:**

Summary of Reviews:

The paper introduces HEM (Ego-Neighborhood Disentangled Encoder with Homophily-aware Environment Mixup), a framework for tackling graph anomaly detection (GAD) under structural distribution shifts caused by selection bias. It addresses a critical problem in real-world GAD applications, such as fraud detection and bot identification. By disentangling node features from structural information and employing a homophily-aware mixup module to generate robust training environments, HEM improves generalization to out-of-distribution graphs. Experiments demonstrate its superiority over various baselines.

Strengths:
-- Novelty: Combines disentangled encoders with an adversarial mixup module, offering an efficient and innovative solution.
-- Comprehensive Evaluation: Outperforms 13 baselines across multiple datasets with strong ablation studies validating its components.
-- Efficiency: The mixup mechanism dynamically re-weights edges instead of creating multiple graph copies, saving memory.
-- Clarity: Well-written and structured, with clear problem formulation and solution explanation.

Weaknesses raised in concerns of reviewers:
-- Lack of Qualitative Analysis: No visualization or deeper analysis of the "invariant patterns" learned by the disentangled embeddings.
-- Complexity: The framework involves multiple modules and loss functions, which could be more intuitively explained.
-- Generalizability: The method relies on a specific GNN (BWGNN); its benefits with other GNNs (e.g., GCN, GAT) are not tested and explained.
-- Outdated Baselines: Some recent GAD and OOD detection frameworks are missing in comparisons (e.g., energy-based models).
-- Dependence on Hyperparameters: The performance depends heavily on the tuning of certain parameters (e.g., ρ).

Conclusion:
HEM presents an innovative and effective approach to GAD under structural distribution shifts, with strong experimental results and practical design. Concerns from reviewers include broader baseline comparisons, qualitative insights, and validation of its generalizability across different GNN architectures. After the rebuttal period, such concerns are well addressed and reviewers are mostly satisfied. Therefore, publication is recommended had the authors include all the discussions and revisions in the final version.